# Associations of Clinical and Dosimetric Parameters with Urinary Toxicities after Prostate Brachytherapy: A Long-Term Single-Institution Experience

Masaya Ito [1,2], Chiyoko Makita [2,*], Takayuki Mori [2], Hirota Takano [2], Tomoyasu Kumano [2], Masayuki Matsuo [2], Koji Iinuma [3], Makoto Kawase [3], Keita Nakane [3], Masahiro Nakano [4] and Takuya Koie [3]

1   Department of Radiation Oncology, Gifu Takayama Red Cross Hospital, 3-11, Tenmancho, Takayama City 500-8717, Gifu, Japan; z2111006@edu.gifu-u.ac.jp
2   Department of Radiology, Gifu University Hospital, 1-1, Yanagido, Gifu City 500-1194, Gifu, Japan; mori_rad@gifu-u.ac.jp (T.M.); htakano@gifu-u.ac.jp (H.T.); tkumano@gifu-u.ac.jp (T.K.); matsuo_m@gifu-u.ac.jp (M.M.)
3   Department of Urology, Gifu University Hospital, 1-1, Yanagido, Gifu City 500-1194, Gifu, Japan; kiinuma@gifu-u.ac.jp (K.I.); buki2121@gifu-u.ac.jp (M.K.); keiaco@gifu-u.ac.jp (K.N.); goodwin@gifu-u.ac.jp (T.K.)
4   Department of Urology, Gifu Prefectural General Medical Center, 4-6-1, Noisshiki, Gifu City 500-8717, Gifu, Japan; mnakano@gifu-hp.jp
*   Correspondence: makita_c@gifu-u.ac.jp; Tel.: +81-58-230-6000

**Abstract:** To examine the association of clinical, treatment, and dose parameters with late urinary toxicity after low-dose-rate brachytherapy (LDR-BT) for prostate cancer, we retrospectively studied patients with prostate cancer who underwent LDR-BT from January 2007 through December 2016. Urinary toxicity was assessed using the International Prostate Symptom Score (IPSS) and Overactive Bladder (OAB) Symptom Score (OABSS). Severe and moderate lower urinary tract symptoms (LUTS) were defined as IPSS $\geq 20$ and $\geq 8$, respectively; OAB was defined as a nocturnal frequency of $\geq 2$ and a total OABSS of $\geq 3$. In total, 203 patients (median age: 66 years) were included, with a mean follow-up of 8.4 years after treatment. The IPSS and OABSS worsened after 3 months of treatment; these scores improved to pretreatment levels after 18–36 months in most patients. Patients with a higher baseline IPSS and OABSS had a higher frequency of moderate and severe LUTS and OAB at 24 and 60 months, respectively. LUTS and OAB at 24 and 60 months were not correlated with the dosimetric factors of LDR-BT. Although the rate of long-term urinary toxicities assessed using IPSS and OABSS was low, the baseline scores were related to long-term function. Refining patient selection may further reduce long-term urinary toxicity.

**Keywords:** prostate cancer; brachytherapy; lower urinary tract symptoms; international prostate symptom score (IPSS); overactive bladder symptom score (OABSS)

## 1. Introduction

Prostate cancer is the second most common cancer worldwide [1]. In Japan, 95,600 men were diagnosed with prostate cancer in 2020 [2]. Treatment options for localized prostate cancer include surgery, external-beam radiation therapy, and brachytherapy, which has shown excellent results in Japan with a 10-year cause-specific survival rate approaching 98% [2]. Regarding surgery, robot-assisted surgery is becoming more widespread, and external-beam radiation therapy, including stereotactic radiotherapy, is beginning to be widely used in oligodose fractionation. There are also options for brachytherapy with both low and high dose rates. Brachytherapy is a method in which a radiation seed is inserted into the prostate gland and provides local irradiation; brachytherapy requires a shorter hospital stay and less physical exertion compared to that of surgery. High-dose rate brachytherapy uses iridium-192 as a radiation-sealed small source that is implanted within

the prostate for a short period. While treatment options for localized prostate cancer are diversifying, brachytherapy monotherapy with an iodine-125 radioactive sealed source is an established method that achieves a high rate of biochemical and clinical control in low- or intermediate-risk patients who are prescribed doses exceeding 140 Gy, and it has been used in Japan since 2003 [3–5]. One advantage of brachytherapy is that, in terms of tumor biology, it can deliver higher doses to the prostate gland compared to that of external radiation therapy. Recent advances in low-dose-rate brachytherapy (LDR-BT) and the placement of a spacer on the rectal side have made it possible to deliver high doses to the target volume while limiting irradiation of adjacent organs at risk.

Because of the excellent long-term overall survival rates achieved using current therapies, long-term toxicity should be carefully considered in light of quality of life after treatment. While treatment of prostate cancer has increased the number of long-term survivors, there are reports of impaired physical and psychological quality of life [6]. For instance, early urinary toxicity (frequent urination, urinary retention, urinary urgency, and urinary incontinence) is common in LDR-BT with iodine-125. Several reports have demonstrated a relationship between symptoms of dysuria and clinical and dosimetric factors [7–14]. Furthermore, there have been reports [15] regarding late urinary dysfunction related to long-term quality of life after LDR-BT, but few studies have examined the relationship between urinary dysfunction and dose factors. Accordingly, the incidence of late voiding symptoms more than 5 years after treatment remains unknown.

The International Prostate Symptom Score (IPSS) as defined by the American Urological Association is the most common patient-completed questionnaire for urinary symptoms. It measures symptom relief after prostatectomy for benign prostatic hyperplasia [16] but is now also used to track symptoms before and after treatment for prostate cancer. In addition, the Overactive Bladder Symptom Score (OABSS) was designed to quantify overactive bladder (OAB) symptoms; the OABSS has a maximum score of 15 points and is more focused on urgency and urge urinary incontinence than frequency. Both the IPSS and OABSS are useful tools for assessing urinary symptoms after brachytherapy [16], but few studies have examined late voiding dysfunction using both IPSS and OABSS. We retrospectively reviewed our institution's experience using IPSS and OABSS to assess clinical, treatment, and dose factor associations in patients with prostate cancer followed for at least 60 months after treatment with LDR-BT. The purpose of this study was to determine the predictive parameters of long-term uremia at 24 and 60 months after LDR-BT.

## 2. Materials and Methods

### 2.1. Patient Population

From January 2004 to December 2016, we evaluated patients with prostate cancer who underwent LDR-BT monotherapy at Gifu University Hospital, Japan. All patients were clinically diagnosed with localized prostate cancer (cT1c-2bN0M0) and classified as low- or intermediate-risk according to the classification proposed by D'Amico et al. Patients with a history of transurethral resection and/or with a maximum flow rate (Qmax) of <10 mL/s based on uroflowmetric assessment were excluded from the indication for LDR-BT. Other indication criteria included an upper limit of prostate volume of 50 mL or less at the time of preplanning and the absence of tumor or inflammatory disease in the rectum (confirmed by prior examination including medical examination and endoscopy). We retrospectively investigated patients who had been followed up for ≥60 months.

### 2.2. LDR-BT

Patients with low-risk prostate cancer with a pretreatment prostate volume (PV) > 50 mL received neoadjuvant androgen deprivation therapy (NADT) for at least 3 months before treatment. Patients with intermediate-risk prostate cancer received NADT for at least 6 months. Patients received iodine-125 sources (Oncoseed, Hihon Medyphysics, Tokyo, Japan) using a real-time transrectal ultrasound-guided transperineal technique [17]. In all cases, iodine-125 seeds were implanted after preplanning using a Mick applicator (Mick

Radio-Nuclear Instruments, Bronx, NY, USA) or ProLink delivery system (CR Bard, Inc., Murray Hill, NJ, USA). The prescribed minimum peripheral dose was 145 Gy. Treatment planning and post-implant dose assessment were performed using an updated American Medical Association Task Group 43 format and Variseed version 7.1 (Varian Medical Systems, Palo Alto, CA, USA). Dose constraints for the organs at risk, particularly the rectum and urethra, were defined. The volume of the rectum receiving 100% of the prescribed dose was constrained to less than 1 cm$^3$. The dose received by 30% of the urethra (UD30) was constrained to <125%. UD10 was constrained to less than 150%.

Our recommended plan included minimizing the dose to both the urethra and rectum.

Patients were routinely administered $\alpha$-1 blockers after LDR-BT to reduce the risk of acute urinary symptoms.

Post-planning was performed 1 month after LDR-BT using computed tomography (CT) and magnetic resonance imaging (MRI). CT was performed using a CT scanner (LightSpeed Ultra 16/Discovery CT 750 HD; GE Healthcare, Milwaukee, WI, USA) with 16 or 64 detector arrays. MRI was performed using a 5-channel SENSE cardiac coil with easy breathing, a 3 mm slice thickness, and no cross gap (Intera Achieva 1.5 T/Intra-Achieva Nova Dual 1.5 T Pulsar; Philips Medical Systems, Eindhoven, The Netherlands).

### 2.3. Follow-Up

The patients were examined and assessed using the IPSS and OABSS before LDR-BT; at 1, 3, 6, 9, and 12 months; and annually thereafter up to 120 months after LDR-BT.

### 2.4. Outcome Measures

The baseline patient characteristics included age, clinical T stage, initial prostate-specific antigen level, Gleason grade group, initial D'Amico risk group stratification, and pretreatment IPSS and OABSS. Treatment parameters included the use of NADT and the numbers of needles punctured and seeds inserted. Dosimetric parameters included the minimal dose delivered to 90% of the prostate volume (PD90), the percentage of prostate volume covered by 100% of the prescription dose (PV100), and the minimal dose received by 5% of the urethra (UD5). UD10 for 10%, UD30 for 30%, and UD90 for 90% of the urethra were defined in the same manner. Uroflowmetry (Qmax) and prostate volume on the treatment day (Intra_PV) were also considered in the analysis.

Moderate lower urinary tract symptoms (LUTS) were defined as IPSS $\geq$ 8, and severe symptoms were defined as IPSS $\geq$ 20.

The OABSS questionnaire contained questions that addressed four symptoms of an OAB: daytime frequency, nighttime frequency, urgency, and urge incontinence. OAB was defined as >2 for nighttime frequency and >3 for total OABSS (score from 0 to 15).

### 2.5. Statistical Analyses

Differences in variables between the groups were assessed via univariate analysis using the Wilcoxon test, $\chi^2$-test, Mann–Whitney $U$ test, and multivariable analysis with logistic regression analysis. The variance inflation factor (VIF) was used to measure the degree of multicollinearity or collinearity in the regression model. The VIF was confirmed to be <2 because of the absence of strong multicollinearity. Statistical significance was set at $p < 0.05$. All statistical analyses were performed using R software (version 3.3.2; R Foundation for Statistical Computing, Vienna, Austria).

### 2.6. Ethical Considerations

The study protocol was approved by the ethics committee of Gifu University Hospital (permission number 22-018). A retrospective chart review was performed for patients who underwent LDR-BT for localized prostate cancer.

### 3. Results

A total of 203 patients (median age: 66 years) were treated with LDR-BT, with a mean of 8.4 years of post-treatment follow-up. NADT was administered to 159 patients. The median prostate volume was 25.4 cm$^3$. A median of 75 seeds (range: 35–108) were implanted with a median number of 22 needles (range: 13–35). Table 1 presents the characteristics of the entire cohort.

**Table 1.** Baseline patient characteristics.

| Factor | | Total 203 |
|---|---|---|
| Age, years | | 66 (50–78) |
| Gleason score | 6 | 143 (50.7) |
| | 7 | 59 (49.3) |
| PSA | | 6 (1.1–14.7) |
| cTstage | T1c | 165 |
| | T2a | 37 |
| | T2b | 2 |
| NADT | Yes | 159 (78.3) |
| | No | 44 (21.7) |
| IPSS | | 5 (0–25) |
| OABSS | | 2 (0–15) |
| OAB | Yes | 39 (23.4) |
| | No | 128 (76.6) |
| UFM (mL/s) | | 10 (10–49) |
| Intra_PV (mL) | | 25.4 (8.77–46.88) |

Abbreviations: IPSS, international prostate symptom score; NADT, neoadjuvant androgen deprivation therapy; OAB, overactive bladder; OABSS, overactive bladder symptom score; PSA, prostate-specific antigen; UFM, uroflowmetry.

Table 2 shows the treatment and dosimetric characteristics of LDR-BT. The median PD90 was 172 Gy (range: 105–241 Gy). The median UD5, UD10, UD30, and UD90 were 235 Gy (range: 171–400 Gy), 222 Gy (range: 168–332 Gy), 218 Gy (range: 159–319 Gy), and 141 Gy (range: 63–241 Gy), respectively.

**Table 2.** Treatment characteristics of the entire cohort.

| Factor | Median | Range |
|---|---|---|
| Needle | 22 | 13–35 |
| Seed | 75 | 35–108 |
| PD90 (Gy) | 172 | 105–241 |
| PV100 (%) | 96 | 77–100 |
| UD5 (Gy) | 235 | 171–400 |
| UD10 (Gy) | 222 | 168–332 |
| UD30 (Gy) | 218 | 159–319 |
| UD90 (Gy) | 141 | 63–241 |

Abbreviations: PD, prescription dose, PV, prostate volume, UD30, dose received by 30% of the urethra.

The median total score of the baseline IPSS was 5 (range: 0–25). The IPSS worsened after 3 months of treatment (median IPSS: 17) and improved to pretreatment levels at 18–36 months in the majority of patients (Figure 1).

The median total score of the baseline OABSS was 5 (range: 0–15). The total OABSS also increased significantly 3 months after LDR-BT compared with the baseline OABSS and returned to the baseline score at 36 months after LDR-BT (Figure 2).

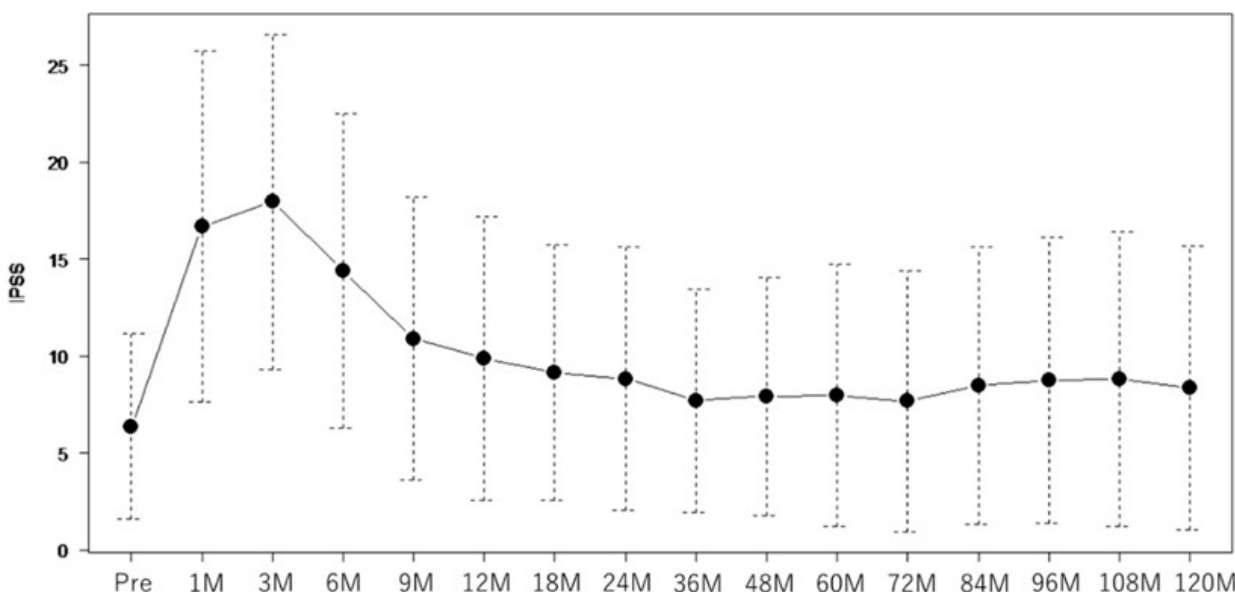

**Figure 1.** Graph of median IPSS score values per period over the 10 years following LDR-BT (all patients). IPSS, international prostate symptom score; LDR-BT, low-dose-rate brachytherapy.

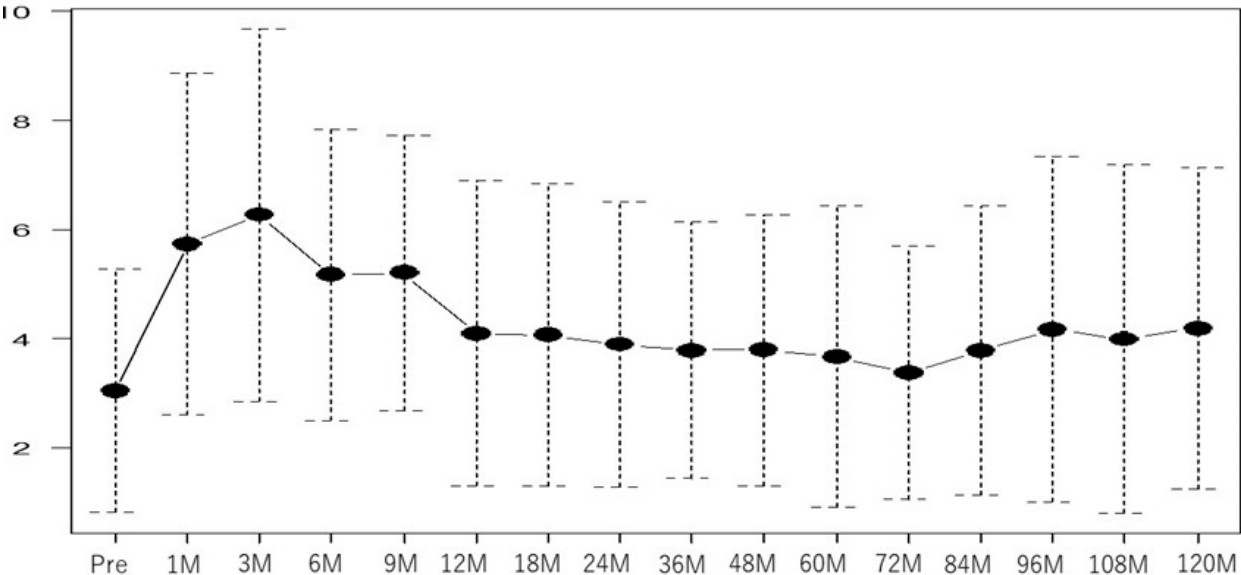

**Figure 2.** Graph of the median OABSS score per period over 10 years following LDR-BT (all patients). OABSS, overactive bladder symptom score; LDR-BT, low-dose-rate brachytherapy.

Table 3 shows the incidence of moderate (≥8 IPSS) and severe (≥20 IPSS) LUTS at 24 and 60 months. The 98 patients who developed moderate LUTS at 24 months had a high baseline IPSS. At 60 months, the baseline IPSS was also associated with moderate and severe LUTS.

Before treatment, 49 patients had been diagnosed with OAB. The pretreatment OAB was significantly correlated with OAB at 24 and 60 months. There was no correlation between LUTS and OAB at 24 and 60 months and the dosimetric factors of LDR-BT. The baseline OABSS was associated with moderate and severe OAB at 24 and 60 months. The baseline uroflowmetry was also associated with OAB at 24 months (Table 4).

**Table 3.** Effect of patient characteristics and treatment parameters on moderate and severe LUTS using IPSS in univariate and multivariable analyses.

| | 2 Years | | | | | | 5 Years | | | | | |
|---|---|---|---|---|---|---|---|---|---|---|---|---|
| | Moderate (*n* = 98) | | | Severe (*n* = 15) | | | Moderate (*n* = 75) | | | Severe (*n* = 12) | | |
| | UVA | MVA | | UVA | MVA | | UVA | MVA | | UVA | MVA | |
| Factor | *p* | OR (95% CI) | *p* | *p* | OR (95% CI) | *p* | *p* | OR (95% CI) | *p* | *p* | OR (95% CI) | *p* |
| Age | 0.61 | | | 0.72 | | | 0.19 | 0.99 (0.91–1.07) | 0.83 | 0.35 | | |
| GS | 0.34 | | | 0.56 | | | 0.53 | | | 1.00 | | |
| PSA | 0.76 | | | 0.42 | | | 0.59 | | | 0.71 | | |
| NADT | 0.22 | | | 0.09 | 0.65 (0.11–3.94) | 0.64 | 0.86 | | | 0.14 | 0.55 (0.21–1.46) | 0.23 |
| Needle | 0.41 | | | 0.71 | | | 0.73 | | | 0.72 | | |
| Seed | 0.11 | 1.02 (1.00–1.04) | 0.02 | 0.20 | 1.02 (0.96–1.07) | 0.56 | 0.59 | | | 0.94 | | |
| Intra_PV | 0.53 | | | 0.37 | | | 0.24 | | | 0.61 | | |
| PD90 | 0.82 | | | 0.37 | | | 0.63 | | | 0.21 | | |
| PV100 | 0.41 | | | 0.75 | | | 0.53 | | | 0.33 | | |
| UD5 | 0.49 | | | 0.25 | | | 0.78 | | | 0.62 | | |
| UD10 | 0.16 | 1.00 (0.99–1.01) | 0.97 | 0.10 | 0.97 (0.94–1.01) | 0.18 | 0.20 | 1.00 (0.98–1.02) | 0.90 | 0.11 | 0.99 (0.98–1.04) | 0.91 |
| UD30 | 0.56 | | | 0.38 | | | 0.71 | | | 0.76 | | |
| UD90 | 0.83 | | | 0.47 | | | 0.84 | | | 0.20 | | |
| Baseline IPSS | <0.01 | 1.3 (1.18–1.42) | <0.01 | 0.08 | 1.12 (0.95–1.31) | 0.17 | <0.01 | 1.19 (1.10–1.29) | <0.01 | 0.02 | 1.18 (1.06–1.31) | <0.01 |
| Baseline UFM | 0.53 | | | 0.72 | | | 0.60 | | | 0.09 | 1.01 (0.94–1.08) | 0.75 |

Abbreviations: IPSS, international prostate symptom score; LDR, low-dose-rate brachytherapy; LUTS, lower urinary tract symptoms.

**Table 4.** Effect of patient characteristics and treatment parameters on OAB using OABSS in univariate and multivariable analyses.

| | OAB at 2 Years (*n* = 60) | | | OAB at 5 Years (*n* = 40) | | |
|---|---|---|---|---|---|---|
| | UVA | MVA | | UVA | MVA | |
| | *p* | OR (95% CI) | *p* | *p* | OR (95% CI) | *p* |
| Age | 0.52 | | | 0.45 | | |
| GS | 0.12 | | | 0.97 | | |
| PSA | 0.71 | | | 0.27 | | |
| NADT | 0.56 | | | 0.19 | | |
| Needle | 0.84 | | | 0.86 | | |
| Seed | 0.16 | | | 0.22 | | |
| Intra_PV | 0.24 | | | 0.47 | | |
| PD90 | 0.42 | | | 0.72 | | |
| PV100 | 0.99 | | | 0.87 | | |
| UD5 | 0.23 | | | 0.75 | | |
| UD10 | 0.19 | 0.99 (0.98–1.01) | 0.78 | 0.40 | | |
| UD30 | 0.40 | | | 0.75 | | |
| UD90 | 0.92 | | | 0.61 | | |
| Baseline OABSS | <0.01 | 1.77 (1.35–2.31) | <0.01 | 0.03 | 2.5 (1.10–5.80) | 0.02 |
| Baseline UFM | 0.02 | 1.11 (1.03–1.20) | <0.01 | 0.61 | | |

Abbreviations: GS, Gleason scale; MVA, multivariable analysis; NADT, neoadjuvant androgen deprivation therapy; OAB, overactive bladder; OABSS, overactive bladder symptoms score; UVA, univariate analysis.

## 4. Discussion

This study described the correlation between clinical and therapeutic factors and LUTS and OAB in patients followed for at least 5 years; the IPSS was highest at 3 months after treatment and improved in the majority of patients at 24 months, when 98 (48%) and 15 (7.3%) patients had LUTS; and at 60 months, when 57 (28%) and 12 (5.9%) patients developed moderate and severe LUTS, respectively. The pretreatment IPSS and

OAB were also significantly correlated with IPSS and OAB at 24 and 60 months upon reobservation; there was no correlation between LUTS and OAB and the LDR-BT dose factor. Previous reports stated that LUTS as measured by the IPSS peaked 7–13 points above the baseline at 1–3 months and that the IPSS gradually decreased to pretransplant levels at 9–18 months [13,18,19]. Similar results were obtained in the present study. In prostate cancer patients with a good long-term prognosis, we observed that LUTS and OAB did not substantially deteriorate even after 2 or 5 years. The OABSS also increased significantly at 3 months after LDR-BT compared to the baseline OABSS but returned to baseline scores 36 months after LDR-BT. Thus, the OABSS showed similar results to those of previous reports concerning urinary symptoms after prostate brachytherapy [16]. In contrast, long-term evaluation is important not only in OABSS but also in IPSS because urethral stricture due to high-dose irradiation at the tip of the urethra may cause dysuria. Several reports indicated that patients with a low baseline IPSS may take longer to improve after an IPSS exacerbation [20–23]. This may be because patients with a high IPSS have a lower difference in changes after symptom exacerbations, while patients with a low IPSS have a lower tolerance for changes and therefore report more symptoms. We believe that this is likely to be the case for the OABSS as well.

Several predictors have been reported for LUTS after LDR-BT. For example, Mischel et al. divided the prostate into three segments (proximal, intermediate, and apical) and investigated the urethral dose to each segment [20]. In this prior report, the possible effect of irradiation to the bladder neck or urethral sphincter on bladder irritation symptoms was also examined, but no association with the proximal urethral dose was noted. Furthermore, UD30 was noted to be associated with acute dysuria, but improvement was observed at 1 year post-irradiation, and with a median follow-up of only 24 months, the effect on late effects is unknown. Acute-phase symptoms have been reported to have no effect on other locoregional urethral doses. For instance, Cary et al. reported that higher doses to the urethral base were associated with an increased number of needles inserted, a higher prostate volume at the time of puncture, and a greater increase in the acute-phase IPSS [21]. In this prior report, a higher urethral base UD50 and PV100 and larger PV predicted a higher maximum increase in the IPSS and a longer IPSS disappearance time. The median follow-up was 44 months; however, late effects on disability were not reported. In addition, James et al. reported that the urethral tip dose was associated with urethral stricture [22]. Specifically, mean doses delivered to the peri-prostatic (mean V150) and apical urethra were significantly higher in patients compared to controls. In addition, the distance from the prostatic apex to the isodose line (approximately 20 mm) was found to be a significant predictor of stricture formation. Urethral stricture often occurs in the globular urethra on the side of the prostatic apex, possibly due to the effect of the dose to the same area. They also reported that the prostatic apex was difficult to source due to surrounding anatomy such as the urethral sphincter and fibrointerstitium; the median follow-up was 45.5 months. Moreover, Steggerda et al. reported that the dose at the bladder neck near the base of the urethra correlated with maximum IPSS [23]. In particular, they found that larger prostate volume, including benign prostatic hyperplasia, was more likely to be associated with a higher dose to the bladder neck and a higher IPSS. Furthermore, Roeloffzen et al. reported that only the dose to the bladder neck was significantly associated with acute urinary retention. The mean D90 to the bladder neck was 65 Gy in acute urinary retention cases versus 56 Gy in controls ($p = 0.016$), and the mean D10 to the bladder neck was 128 Gy versus 107 Gy ($p = 0.018$) [24]. They also stated that the risk of acute urinary retention increased with greater protrusion into the bladder due to benign prostatic hyperplasia. As can be seen in these reports, some dose factors may be predictive of early symptoms after treatment, but there are currently no reports on the relationship between IPSS and OAB and dose factors (such as after 60 months). These reports suggest that irritation to the bladder neck may result not only from physical irritation due to prostatic hypertrophy but also from radiation-induced inflammation and that this irritation may be a significant predisposing factor for dysuria. Notably, Onishi et al. reported that symptoms of both

urethral obstruction as well as those caused by irritation are important in late dysuria [21], and we believe that long-term evaluation by OABSS that considers irritation of the bladder is also important. On the other hand, since urethral stricture due to high-dose irradiation of the urethral tip may cause dysuria, we believe that long-term evaluation via the IPSS as well as the OABSS is important.

In addition, a temporary worsening of urinary status (flare) may be observed during the course of IPSS improvement [23–25] as observed by Cesaretti et al., who reported that flares may persist for a few months and often appear within 5 years post-operatively. Although the number of seeds and needles as well as a patient age <65 years tended to be associated with flares, the association did not achieve statistical significance, including a relationship with the urethral dose.

Therefore, the long-term impact of this study on the IPSS score is likely to be small. In this study, we suspected an association between late voiding dysfunction and the urethral dose. It is noteworthy that several reports suggested that long-term urodystrophy is not high in many patients with prostate cancer [26–29], although this can occur in certain patients 10 years or more after LDR-BT. This was consistent with our results (Figures 1 and 2) and was corroborated by the findings of Keyes et al., who reported a median follow-up of 54.5 months and dose factors for PD90 and PV100 [26]. Moreover, Stone et al. reported that a high radiation dose (including external beam radiation), hypertension, and alcohol consumption were associated with late dysuria in patients with low dysuria before treatment [28]. These reports identified a larger prostate volume and baseline IPSS as factors associated with worse urinary symptoms, but they did not address the association between specific dose factors and the IPSS in late-life dysuria. We attempted to determine the dose parameters that were predictive of long-term urinary symptoms and also found that the baseline IPSS was associated with long-term IPSS scores; however, none of the dose parameters correlated strongly with the IPSS or OABSS, and we did not find a predictive dose parameter for long-term dysuria. Other factors such as the prostate volume [21–24], number of needles at the time of puncture [21], and presence of NADT [21,22] have also been suggested to be associated with the IPSS in previous reports, but we found no association between the IPSS and OABSS in the late phase in this study.

The European Society for Radiotherapy and Oncology/European Association of Urology/European Organisation for Research and Treatment of Cancer's reported guidelines for prostate brachytherapy state that the main parameters for Urestra should be D10 < 150% of the prescribed dose and D30 < 130% of the prescribed dose [30]. Our planning dose constraints were that UD10 should be at least 150% or less and UD30 should be 125% or less. In the present study, we examined the association between late dysuria and dose factors in cases with a longer follow-up period of at least 60 months than previously reported. The results suggested that the urethral dose contributed little to late dysuria if the LDR-BT dose coefficients met the guideline dose constraints and that the risk of treatment-induced worsening of urinary symptoms was very low even after 5 years. Although there are some reports [22,31] that there was no rationale for reducing urethral dose at the expense of prostate dose based on a short-term follow-up, the present results suggested the same for long-term follow-up results. Nonetheless, this study had several limitations. First, this was a single-center study with a small number of cases. Second, this study was retrospective in design. Third, two types of seeds were used in LDR-BT (the Mick applicator and the ProLink delivery system), and these were used by multiple operators. Fourth, the mean follow-up period of this study was 8.4 years, which was insufficient to assess late voiding dysfunction in the long-term prognosis after prostate cancer treatment, and a longer follow-up period is needed. Fifth, this study evaluated LDR-BT alone, and comparisons with other therapies in the long-term perspective are also needed. Sixth, it was not possible to follow up all patients. This was due to factors such as the patients' own evaluations of their conditions after treatment and their inability to visit the hospital.

## 5. Conclusions

The rates of long-term urinary toxicities assessed using IPSS and OABSS were low, and the baseline IPSS and OABSS were related to long-term IPSS and OABSS, respectively. Although the LDR-BT performed in this study complied with the planning dose constraints and showed excellent results at 24 and 60 months, refining patient selection may further reduce toxicity.

**Author Contributions:** Conceptualization, M.I. and C.M.; methodology, M.I., T.M., H.T., T.K. (Tomoyasu Kumano), C.M. and M.M.; investigation, M.I. and C.M.; resources, M.I., T.M, H.T., K.I., M.K., K.N., M.N., T.K. (Takuya Koie) and M.M.; data curation, M.I. and C.M.; writing—original draft preparation, M.I. and C.M.; writing—review and editing, C.M, T.K. (Takuya Koie) and M.M.; visualization, C.M.; supervision, M.M. All authors have read and agreed to the published version of the manuscript.

**Funding:** This research received no external funding.

**Institutional Review Board Statement:** The study protocol was approved by the ethics committee of Gifu University Hospital (permission number 22-018). A retrospective chart review was performed for patients who underwent LDR-BT for localized prostate cancer.

**Informed Consent Statement:** In accordance with the ethical guidelines established by Japan, in non-invasive observational research, patient consent is confirmed by opt-out, in which information about the conduct of the research, including the purpose of the research, is notified or disclosed.

**Data Availability Statement:** The data presented in this study are available upon request from the corresponding author. The data are not publicly available for privacy and ethical reasons.

**Conflicts of Interest:** The authors declare no conflict of interest.

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
