# Peer review of "Associations of Clinical and Dosimetric Parameters with Urinary Toxicities after Prostate Brachytherapy: A Long-Term Single-Institution Experience"

_curroncol, doi:10.3390/curroncol30060426_

Round 1
Reviewer 1 Report (Previous Reviewer 2)
Please revise your paper according to the comments in the attached file.

Author Response
Point 1: Several issues should be addressed in this reviewer`s opinion before potential publication of
this study:
Abstract: raw 26-28 - the statement contradicts the results of the study.
response: we revise Abstract to match the text content.
Material and Methods - raw 70-72 - unclear sentence, must be re-written; needs much more info on exclusion criteria for LDR-BT - critical for the interpretation of the results -
INFO - e.g. upper limit of prostate volume for LDR-BT, rectal pathology, IBD, etc
STILL NEED ADDITIONAL
Response: we add more info on exclusion criteria for LDR-BT.
Raw 109 of the revised abstract - ureter?
Response: we fix the word.
raw 109-100 significant unclarity regarding two terms:
- uroflowmetry - very broad term - needs additional info - Qmax, Qave, cut-off values, use as continuous variable? - still unclear
Response: we add explanation of the word “uroflowmetry”.

Round 2
Reviewer 1 Report (Previous Reviewer 2)
the authors have sufficiently addressed the raised issues
This manuscript is a resubmission of an earlier submission. The following is a list of the peer review reports and author responses from that submission.
Round 1
Reviewer 1 Report
This study is aimed at the evaluation of long-term urinary toxicities after low-dose rate prostate cancer brachytherapy. Urinary toxicity in 203 patients was assessed using the international prostate symptom and overactive bladder symptom scores.
Findings are original, as limited data is available on late urinary toxicity after low-dose rate prostate cancer brachytherapy. The reported findings are expected and are in line with the short-term findings reported after prostate cancer brachytherapy. Overall results are not entirely novel but worthwhile to consider.
This paper provides clinical evidence to support an expected but not previously reported observation.
Clinical methodologies are appropriate. However, consulting a biostatistician reviewer might be required to judge the suitability of the statistical approach.
Conclusions are consistent with data, and results are adequately discussed.
The references are appropriate.
No additional comments on the tables and figures.
Author Response
Thank you for review.
Reviewer 2 Report
the authors are presenting interesting study focused on prognostic factors for long-term urinary toxicities after LDR-brachy. The main strength of the study is the long-term follow-up, which is mandatory with the excellent oncological results achieved in early prostate cancer treatment with LDR-BT. Most of the literature on urinary toxicity is focused on the early- and mid-term results.
Several issues should be addressed in this reviewer`s opinion before potential publication of this study:
Abstract: raw 26-28 - the statement contradicts the results of the study -
Patients with a higher baseline IPSS and OAB had a lower frequency of moderate and severe LUTS 27 and OAB at 24 and 60 months, respectively
Introduction - comprehensive and clear paragraph, nicely defining the purpose and scientific soundness of the study
Material and Methods - raw 70-72 - unclear sentence, must be re-written; needs much more info on exclusion criteria for LDR-BT - critical for the interpretation of the results
raw 109-100 significant unclarity regarding two terms:
- uroflowmetry - very broad term - needs additional info - Qmax, Qave, cut-off values, use as continuous variable?
- intra PV (prostate volume?) - significant discrepancy - latter in the text it is stated that median prostate volume is 25,4 ml, while in table one intra PV is stated as 117 (69-174) ml - this is impossible in LDR-BT in this reviewer`s opinion. Could it be the authors are referring to post-void residual (PVR?)
raw 136-137 - PSA abbreviation - membrane? a typo?
raw 159 large number of seeds - cut-off? continuous variable?
raw 167 - 168 - discrepancy in the sentence - OABSS correlates with OAB frequency in table 4, not LUTS
Discussion and Conclusion - nicely written paragraphs comparing authors results with the literature and adequately substantiate their results
the references are outdated and should be re-checked
Regarding all the aforementioned, my recommendation is to re-assess this manuscript for publication after satisfactory comments and appropriate modification by the authors on the abovementioned issues.
